# Identification of the Angle Errors of the LED Parallel-Light Module in PCB Exposure Device by Using Neural Network Learning Algorithms

**Chi-Feng Chen [1,*], Jian-Rong Chen [2] and Ting-Yu Chen [3]**

[1] Department of Mechanical Engineering and Institute of Opto-Mechatronics Engineering, National Central University, Taoyuan City 32001, Taiwan

[2] Institute of Opto-Mechatronics Engineering, National Central University, Taoyuan City 32001, Taiwan

[3] Institute of Molecular Medicine and Bioengineering, National Yang Ming Chiao Tung University, Hsinch 30010, Taiwan

[*] Correspondence: ccf@cc.ncu.edu.tw; Tel.: +886-3-426-7308

**Abstract:** For the smart manufacturing development of printed-circuit-board (PCB) exposure devices, the LED parallel-light (LPL) module is investigated and the angle errors of those LPL units are identified by neural network learning algorithms. At present, in PCB manufacturing, most circuit boards use photoresist covering etching. After exposure and development, unwanted copper foil is etched and removed to make circuit boards. The exposure process is its key process, and the equipment used in this process is an exposure machine. The LPL unit is designed and the LPL exposure module is searched under the principle of higher irradiance uniformity. The learning data of supervised learning for the convolutional neural network (CNN) include a 2D irradiance distribution image constructed by the ray tracing simulation tool. In these supervised learning data, all units of LPL-EM are randomly added with a self-specific angle error. By using Fast Region-based CNN, the identification of the multi-LPL module with the specific errors of inclination and azimuth angle is verified. Those results preliminarily illustrate that supervised learning techniques should be able to help identify the errors of inclination and azimuth angle for the single LPL unit and multi-light module of PCB exposure devices. In other words, this technology should serve as a reference for the development of the PCB exposure process towards smart manufacturing.

**Keywords:** smart manufacturing; convolutional neural network; Fast R-CNN; PCB exposure device

## 1. Introduction

With the development of information technology, manufacturing technology is gradually developing towards the direction of so-called smart manufacturing that incorporates information technology and artificial intelligence (AI) [1–3]. Smart manufacturing can help to increase the flexibility and efficiency of factories, increase manufacturing yield or quality, and reduce the use of manpower and energy. Moreover, it is also more in line with the requirements of environmental sustainable development [4]. It is well known that the concept of sustainable manufacturing has gradually become one of the driving forces behind the development of the manufacturing industry. To achieve the smart manufacturing goals, virtualization technology of physical entities [5] and machine learning algorithms are imported [6–8]. To support intelligence learning or deep learning [9–13], several learning algorithms that allow computers to learn automatically are proposed, such as backpropagation (BP), convolutional neural network (CNN), and recurrent neural network (RNN), and applied in many fields, such as computer vision, natural language processing, speech recognition, handwriting recognition, biometric identification and medical diagnosis. BP learning uses the chain rule to calculate the gradient of the loss function with the weights of each layer of the network as variables, and updates the weights to minimize the loss

function [12]. CNN is a feedforward neural network [12,13]. Due to the convolutional layer and pooling layer in its architecture, it has the benefit of strengthening pattern recognition and the relationship between adjacent data. Based on its unique advantages in pattern recognition, CNN has achieved good results in the application of image and speech recognition. To enhance the detection performance for the recognizing task of semantic segmentation, a region-based CNN (R-CNN) method is proposed [14,15]. Compared to the best previous results achieved with the PASCAL visual object casses (VOC) 2012, the R-CNN algorithm with the rich hierarchy of image features is proven to show 30% relative improvement [14]. To increase learning speed and enhance detection accuracy for the object detection, the Fast Region-based convolutional network method (Fast R-CNN) that employs several innovations is proposed [16,17]. It is shown that, compared to R-CNN and the spatial pyramid pooling network (SPPnet), Fast R-CNN's speed of training and testing is several times faster, and has higher accuracy [16]. The Faster RCNN applied in face detection has impressive results [18]. RNN has the characteristics of memory, parameter sharing and Turing completeness, and is a kind of neural network specially used to solve time-related problems [9,10]. RNN can efficiently learn nonlinear features of sequences, so it has achieved good results in speech recognition, language modeling, and machine translation applications.

In printed-circuit-board (PCB) manufacturing systems, the UV exposure process is one of the key processes [19,20]. UV lamps used in traditional UV exposure equipment are mercury-containing products. According to the spirit and development of the Minamata Convention on Mercury [21], which is an international convention to comprehensively regulate mercury, the future of mercury-containing UV lamps will surely be regulated. Under the influence of sustainable manufacturing and related environmental protection conventions, the UV-LED exposure machine is expected to gradually replace traditional exposure equipment and become the mainstream. The basic requirements for exposure of traditional PCB parallel-light exposure machines are as follows: 1. the irradiance is at least 20 mW/cm$^2$, 2. the irradiance uniformity is 90%, 3. the light collimation includes the parallel-light half angle and inclination angle within at least $4°$, and 4. the wavelength is preferably 365 nm, 405 nm, and 436 nm. Looking at the current status of UV-LED technology, for UV-LED parallel-light exposure machines to fully replace traditional UV parallel-light exposure machines, there are still some technical challenges.

In this paper, based on the demands of realizing smart manufacturing for the LED parallel-light (LPL) module of PCB exposure devices, we will investigate the LPL-EM's angle errors in all LPL units and identify the angle errors of those LPL units, due to manufacturing and assembly errors, by CNN learning algorithms (CNNLA). First, an LPL unit is designed and virtualized, and identified by the CNN and Fast R-CNN learning algorithms. The learning data of supervised learning for the CNN include a 2D irradiance distribution image built by the ray tracing simulation tool, when it is assumed that all units of the LPL-EM randomly have their self-specific angle errors. The technology of the automatic identification of the azimuth and inclination angle errors of the LPL-EM by using Fast R-CNN with Haar-like features is realized and verified. Obviously, the supervised learning technology can be effective at identifying the single LPL unit and multi-LPL-EM of PCB exposure devices. Based on those results, it is possible to further understand the exposure devices of the PCB exposure process in detail and analyze their manufacturing problems, thereby realizing the goal of smart manufacturing.

## 2. Design of the LPL Unit and LPL Exposure Module

### 2.1. Design of the LPL Unit

For UV exposure machines in PCB production, higher light parallelism and higher irradiance uniformity (IU) on the exposure working plane mean that they have higher

efficiency performance. Here, the IU is defined as the ratio of the minimum irradiance to the maximum irradiance among all irradiated pixels and can be written as follows:

$$IU = \frac{E_{min}}{E_{max}} \times 100\ \%, \tag{1}$$

where $E_{min}$ and $E_{max}$ represent the minimum and maximum values of irradiance among all the irradiated pixels, respectively. The LPL exposure module (LPL-EM) is composed of LPL units arranged in a hexagonal shape. The LPL unit mainly contains an LED chip, metal core PCB (MCPCB) substrate, and total internal reflection (TIR) lens. Their configuration relationship is as follows: the LED chip and TIR lens are bonded on a MCPCB substrate and the LED chip is positioned in the center accommodation space of the TIR lens. The optical collimation degree of the LPL-EM is directly related to the optical collimation degree design of the unit, the manufacture and assembly of the unit. Therefore, it is necessary to design and manufacture the LPL unit with higher light parallelism and further optimize all unit positions of LPL-EM under the target of higher IU. In addition, it must be noted that UV is relatively harmful to the human body, and the wavelength factor has little effect on the validity of ray tracing simulation. Therefore, under the premise of considering the convenience and reliability of manufacturing, and the safety and convenience of the experiment, we use white LEDs to replace the UV-LEDs in this study. The LPL unit and LPL-EM is designed by the ray tracing simulation tool and the normalized radiant intensity and normalized irradiance distribution of the designed LPL unit on the exposure working plane are shown in Figure 1, where the distance between the exposure working plane and the MCPCB substrate of the LPL-EM is 400 mm. For the radiant intensity of the designed LPL unit, half of the full width at half maximum (FWHM) is about 3.8°. One can observe from Figure 1b that the magnitude of the normalized irradiance distribution is represented by the image color, and the corresponding relationship between the color and the normalized irradiance magnitude can be observed from the color contrast chart.

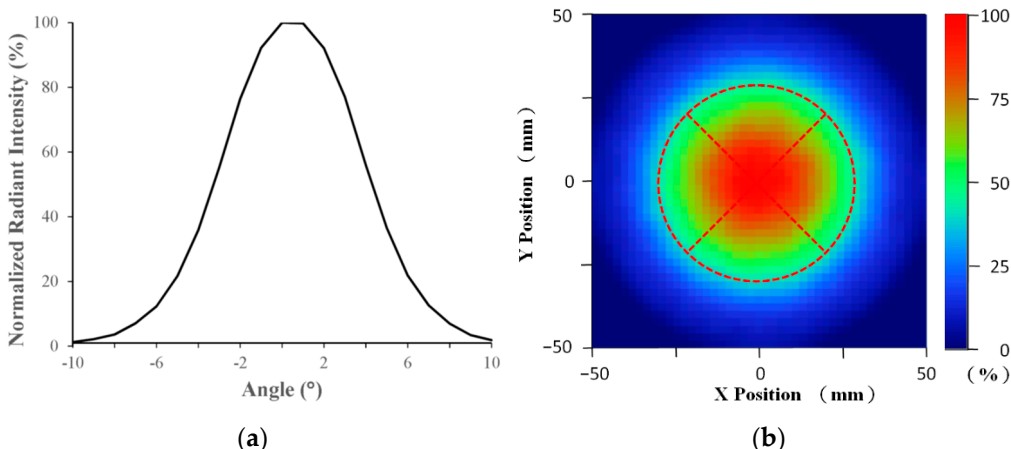

(a)  (b)

**Figure 1.** (**a**) Angle distribution and (**b**) irradiance distribution of the designed LPL unit on the exposure working plane.

In the real physical world, manufacturing and assembly errors are inevitable. Those errors can cause the radiant intensity angular distribution of each LPL unit, called the light distribution reference direction, to deviate from the ideal surface normal direction and the deviation is described by the inclination angle and azimuth angle. Here, the inclination angle refers to the angle between the light distribution reference direction and the ideal surface's normal, and the azimuth angle refers to the angle between the light distribution reference direction and the positive x-axis. Four irradiance distributions of the LPL units with specified deviations are shown in Figure 2, where highlighting marking is added to emphasize the variation of each major irradiance distribution. As can be

seen from Figure 2, the changes in these highlighted marks are conditions that satisfy the original assumptions. To verify the validity of the ray tracing simulation, the design sample involves trial manufacturing. The photos of the measurement architecture, the irradiance distribution without external error, and the irradiance distribution with the external angle error of 2° for the experimental results of the LPL unit are shown in Figure 3. It can be observed that the test sample has a good irradiance distribution and the experimental results are similar to the simulation results.

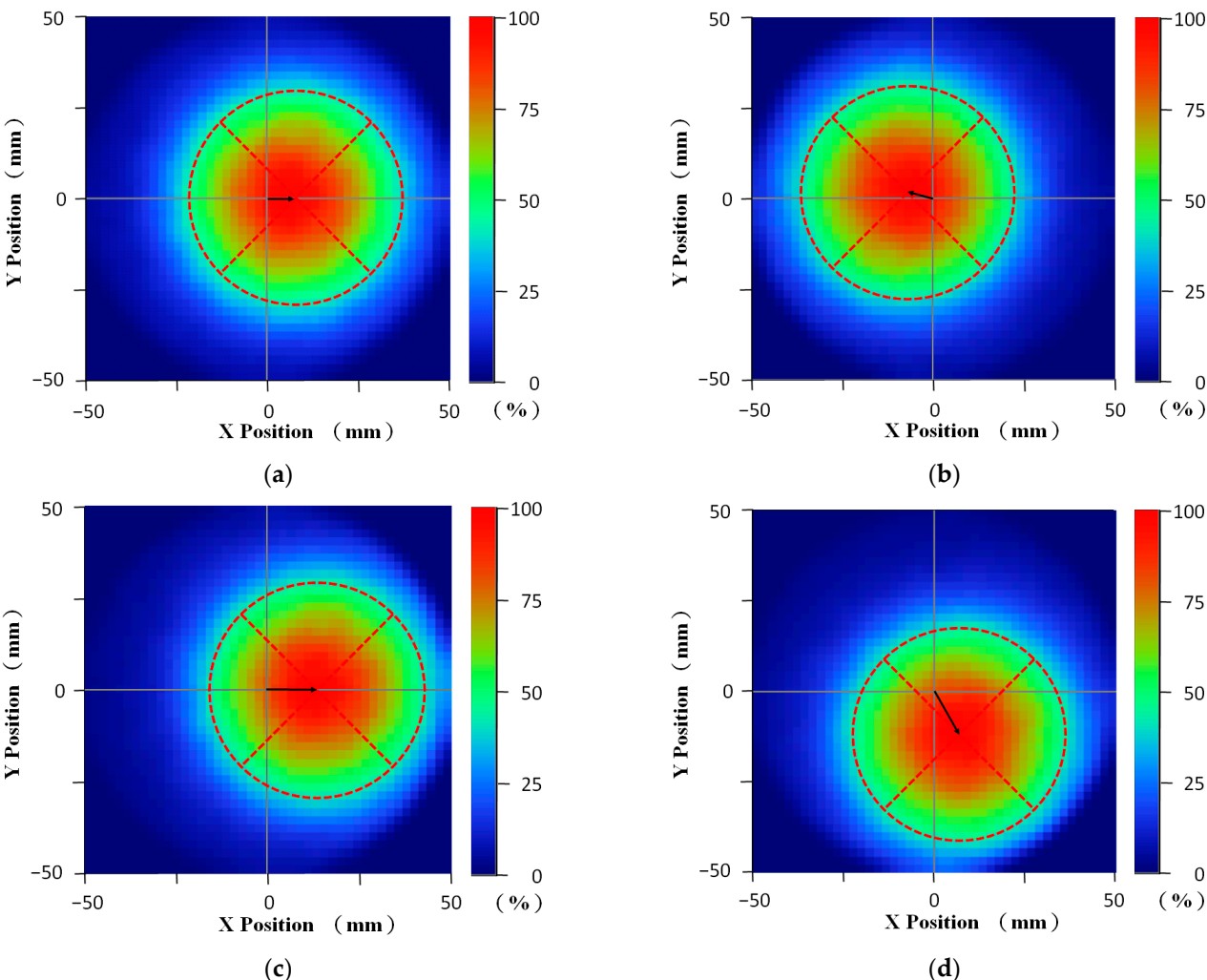

**Figure 2.** Four irradiance distributions of the LPL units with specified deviations on the exposure working plane: (**a**) inclination angle 1° and azimuth angle 0°, (**b**) inclination angle 1° and azimuth angle 165°, (**c**) inclination angle 2° and azimuth angle 0°, and (**d**) inclination angle 2° and azimuth angle 300°.

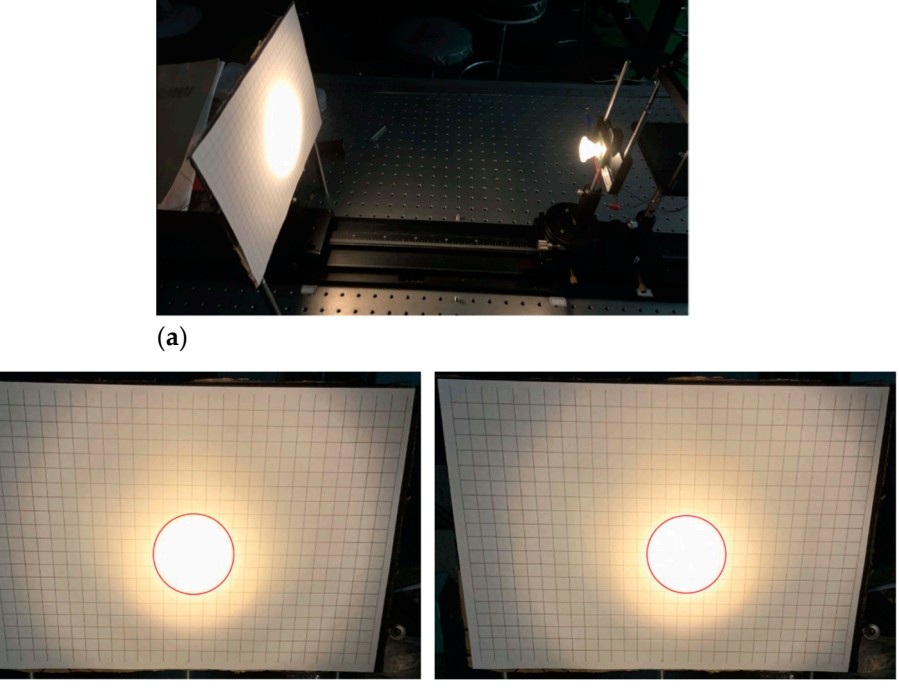

**Figure 3.** Photos of (**a**) the measurement architecture, (**b**) the irradiance distribution without external error, and (**c**) the irradiance distribution with the external angle error of 2° for the experimental results of the LPL unit.

### 2.2. Design of the LPL Exposure Module

Next, to meet the target of higher IU, the distance between the centers of two adjacent LPL units, called the unit pitch, is further analyzed by the ray tracing simulation tool [22]. A simple LPL-EM, consisting of 23 LPL units arranged in a hexagonal shape, is considered and shown in Figure 4. It can be observed that these 23 units can be divided into the following two categories: one is called the edge unit located at the boundary, and the other is called the intermediate unit that is not located at the boundary. There are 16 edge units in total, and the corresponding exposure area has an obvious edge effect. There are seven intermediate units, which are composed of six units at the top corners of a regular hexagon and a unit at the center of the hexagon. The edge effect of the corresponding exposure area is small. This is why the 23 units in this hexagonal configuration are called the simple LPL-EM. The simple module is used to study the optical quality of LPL-EM, which has the characteristics of representativeness, simplicity, symmetry, and rapidity. Figure 5 shows the variation in the IU with LPL unit pitch in the area that is less affected by the edge effect. It can be observed that when the LPL unit pitch is 51.4 mm, the higher IU is obtained about (92.9%). Therefore, the LPL unit pitch is taken as 51.4 mm. In Figure 6a, one can observe that there are obvious edge effects around the irradiation region. As mentioned above, the IU of the parallel-light exposure machine is 90% and most of the IU in those peripheral areas obviously cannot meet this standard. Therefore, the effective exposure region of the LPL-EM must be explored. In addition, since some errors in manufacturing and assembly cannot be completely avoided, these errors will lead to a consequent drop in IU, and higher tolerances in manufacturing and assembly help to reduce the production of finished products. Therefore, a larger area and a higher IU are set as the exploration goals, and the exploration results shown in Figure 6b show that the area is $150 \times 150$ mm$^2$ and the IU is 92.9%. In Figure 6b, the grids are taken as $25 \times 25$, that is, the pixel size of the grid is $6 \times 6$ mm$^2$. For the IU detection standard commonly used in the industry, the UV irradiation meter is used to regularly measure 25 positions, 5 positions for each row and column, within the set effective exposure area. The diameter of the measuring range of the

irradiance meter is generally greater than 20 mm. Comparing this practice standard with the IU analysis standard for this study, this analysis standard is obviously higher than the practice standard. It is a decision that takes into account both the applicability and accuracy of the research. In addition, in the effective exposure region, the influence of the edge effect is small, and the irradiance field is relatively symmetrical, so the IU is relatively higher.

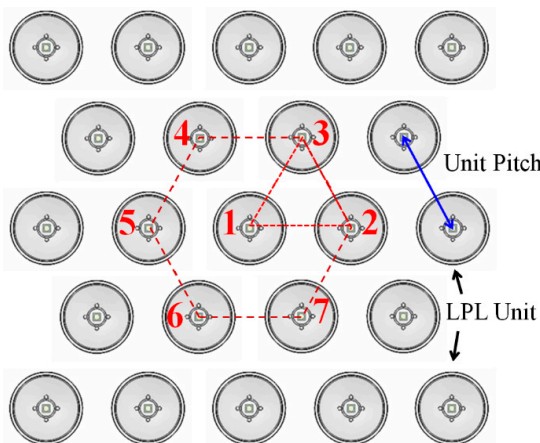

**Figure 4.** Schematic of a simple LPL-EM consisting of 23 LPL units arranged in a hexagonal shape.

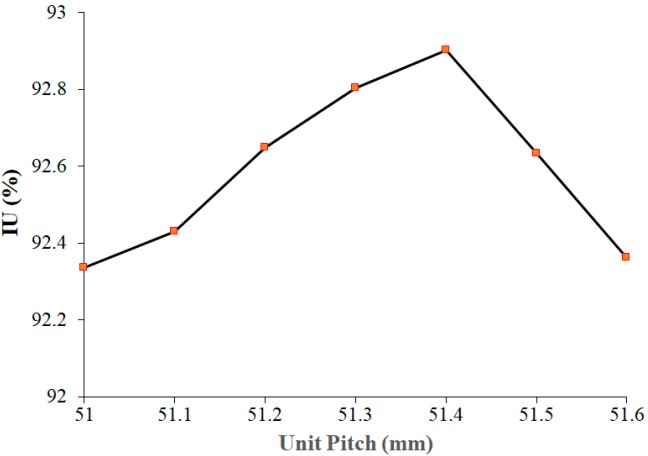

**Figure 5.** Variation in the IU with LPL unit pitch.

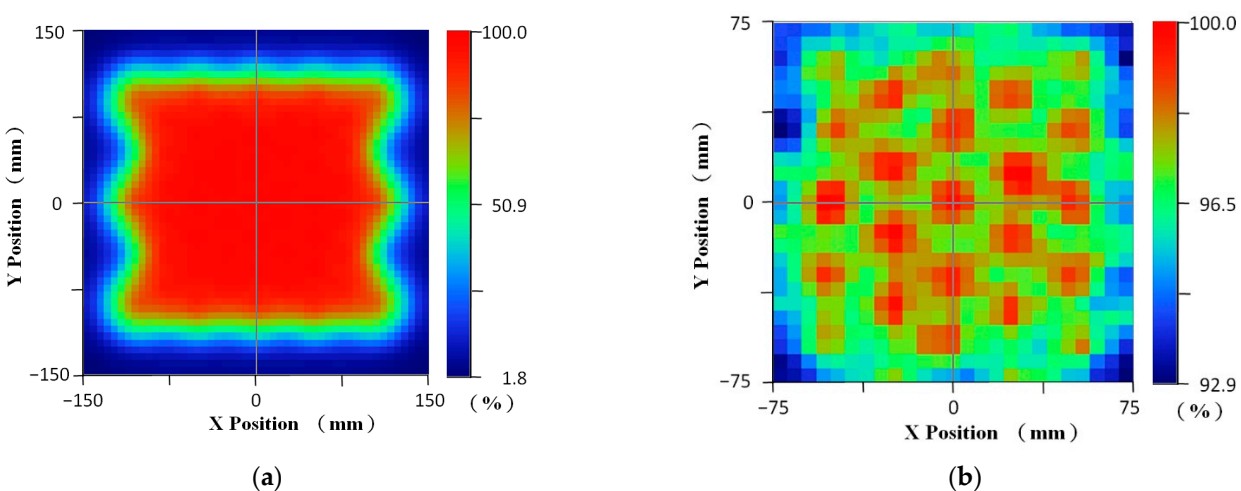

**Figure 6.** Irradiance distributions of the designed LPL-EM on the working plane: (**a**) irradiated region and (**b**) effective exposure region.

### 3. Identification of the Angle Errors of LPL Unit by Applying CNNLA

*3.1. Using CNN*

As the irradiance distribution is a two-dimensional image, the CNNLA is considered for neural network learning. The irradiance distribution of the LPL unit with a given angular error is simulated and obtained by the ray tracing simulation tool and it is used to make learning material for the CNN learning program (CNNLP) based on Keras in Python. The setting of the basic architecture of CNN is as follows: two layers of convolution layers, two layers of pooling layers, and one layer of fully connected layers as learning models. To improve the accuracy and learning efficiency, cross entropy (CE) is taken as a function of computing loss and estimation error, and can be written as follows [9]:

$$\text{CE} = -\sum_{i-1}^{N} p_i(x) \times \log q_i(x), \tag{2}$$

where $N$ is the test set size, $q(x)$ represents the estimated probability of event x occurring on the learning set, and $p(x)$ represents the true sample result. The output value of CE is between 0 and 1. A perfect model would have a log loss of 0. In addition, two convolutional layers, two pooling layers, and one fully connected layer are set as learning models in the CNNLP. The frequency of updating parameters is one batch for every four input data.

First, it is assumed that the inclination angle is 1° and the azimuth angle is randomly selected from 0° to 360°. According to the mechanism of azimuth error generation, this error should be direction-independent, so the probability of occurrence is assumed to be the same at any angle of azimuth from 0° to 360°. The elementary learning objective is to distinguish which quadrant the azimuth angle of the learning sample is in. The total number of input data is 3000, which is divided into two parts, training and testing, and the numbers of training data and testing data are generated by random sampling of the total data in the region according to a preset proportion. Numbers of training data and testing data for first learning epoch are shown in Table 1, where the quadrant is one part of the irradiated plane divided into fourths. The first quadrant is located in the area where both coordinate values are positive, and the other quadrants are given in reverse clockwise order. Following the above data, the first learning epoch is performed. Those data are resampled before each learning session. The learning epochs are set as 10, and the results based on the termination condition, CE < 0.05, are listed in Table 2. It is shown from the learning results that the testing accuracy gradually converges and the testing accuracy reaches more than 80% after the 7th learning epoch. After 10 epochs, the testing accuracy has reached 86.8%.

**Table 1.** Numbers of training data and testing data for first learning epoch.

|  | First Quadrant | Second Quadrant | Third Quadrant | Fourth Quadrant |
|---|---|---|---|---|
| Number of training data | 588 | 613 | 594 | 607 |
| Number of testing data | 146 | 153 | 148 | 151 |
| Total | 734 | 766 | 742 | 758 |

**Table 2.** Accuracy results of 10 learning epochs for the elementary learning objective.

| Number of Epochs | Training Accuracy (%) | Testing Accuracy (%) |
|---|---|---|
| 1 | 77.0 | 76.2 |
| 2 | 78.9 | 77.3 |
| 3 | 80.6 | 78.1 |
| 4 | 81.4 | 80.1 |
| 5 | 81.1 | 79.2 |
| 6 | 81.7 | 79.1 |

**Table 2.** *Cont.*

| Number of Epochs | Training Accuracy (%) | Testing Accuracy (%) |
|:---:|:---:|:---:|
| 7 | 82.9 | 82.0 |
| 8 | 84.0 | 83.1 |
| 9 | 85.1 | 84.9 |
| 10 | 86.7 | 86.8 |

*3.2. Using Fast R-CNN*

Next, a further learning objective is to be able to efficiently classify learning samples within 15° of azimuth. The accuracy results of the training and testing are shown in Table 2. This learning task is clearly a failure. It is because the learning effect is not good when the azimuth angle of the learning sample is close to the boundary region for azimuth classification, that is, this learning method is not suitable for learning tasks with a high proportion of boundary regions. To solve this problem, the Fast R-CNN method is chosen and several feature boxes are set. The individual features of the adjacent feature boxes are strengthened to improve the learning accuracy. The learning conditions are the same as the CNN in Section 3.1. The learning results of 10 epochs are listed in Table 3 and the testing accuracy reaches 78.3% through 10 learning epochs. It is shown that the Fast R-CNN method can effectively resolve irradiance images up to an azimuth angle of 15°. Thus, the automatic identification technology that the automatic identification of LPL units in exposure machines by using Fast R-CNN is realized and can also be used to seek the angle errors of lighting devices, based on their irradiance distribution image. Although the current method can effectively reduce the learning interference caused by the partition boundary effect, the current learning accuracy is still not ideal. The improvement in accuracy is a topic that can be further continued in the future.

**Table 3.** Accuracy results of 10 learning epochs for the advanced learning objective.

| Number of Epochs | Training Accuracy (%) | Testing Accuracy (%) |
|:---:|:---:|:---:|
| 1 | 70.1 | 69.9 |
| 2 | 70.9 | 71.4 |
| 3 | 73.3 | 73.2 |
| 4 | 73 | 75 |
| 5 | 74.8 | 74.6 |
| 6 | 75.1 | 75.3 |
| 7 | 77.6 | 76.1 |
| 8 | 77.3 | 75.8 |
| 9 | 78.3 | 77.4 |
| 10 | 79.1 | 78.3 |

## 4. Identification Angle Errors of the LPL Units in the Simple LPL-EM

*4.1. Description for the Haar-Like Feature Method*

For identifying the angle errors of the LPL units in the simple LPL-EM, the Haar-like feature is imported in the learning algorithm [23]. A rectangular Haar-like feature is defined as the difference between the pixel sums of several regions in the rectangle and can be written as follows:

$$ii(x,y) = \sum_{x' \leq x, y' \leq y} i(x',y'),\qquad(3)$$

where $ii(x,y)$ is the integral image and $i(x',y')$ is the original image. That is, the sum of the pixels in the rectangular area can be represented by the value of the integral image.

Furthermore, to quickly calculate the integral image, the following relational equations are used [23]:

$$s(x,y) = s(x, y-1) + i(x,y), \tag{4}$$

$$ii(x,y) = ii(x-1, y) + s(x,y), \tag{5}$$

where $s(x,y)$ is the cumulative row sum. Using Equations (4) and (5), the integral image can be computed in one pass over the original image. The schematic diagram of the calculation of the Haar-like feature by using the integral image is shown in Figure 7. In Figure 7, the value of the integral image at location 1–4 can represent the sum of the pixels in rectangle A, A+B, A+C, and A+B+C+D, respectively. Therefore, the sum of the pixels within rectangle D can be calculated as 4+1−2−3, that is, the sum of the pixels within rectangle D can be calculated for four array references. Using this method, any rectangular sum in the image of irradiance distribution can be quickly calculated as four array references.

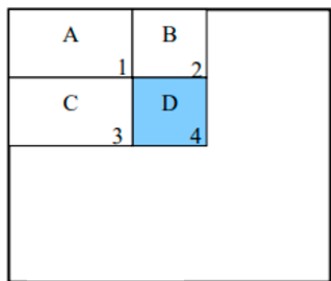

**Figure 7.** Schematic diagram of the calculation of the Haar-like feature by using the integral image.

*4.2. Description of the Feature Classification of the Irradiance Distribution*

In this study, the considered LPL-EM arranged in a hexagonal shape, as shown in Figure 3, can be observed as a composite of regular triangles formed by three light sources. In the real module, each LPL unit has its own angle error. To directly perform feature classification for the LPL-Ems is a very complex and difficult problem. To simplify this problem, the basic analysis model (a regular triangle formed by three LPL units) is considered and shown in Figure 8. It is used to explore the feature classification of the irradiance distribution in the regular triangle area when angle errors exist in those LPL units. For the basic analysis model, the three LPL units have the characteristics of rotational symmetry (rotational symmetry of 60°). One unit is appointed as the first unit and marked as "1", the second unit is marked as "2", and the third unit is marked as "3", as shown in Figure 8. The second and third units are symmetrical about the mirror axis, so their mirror images are the same and are classified into the same category. In addition, based on the learning research results of the LPL unit, the LPL-EM continues to use previous learning tools and learning conditions. The accuracy target is set to 70%. First, the first and third units are fixed, and the effect of angle deviation of the second unit at different angles is discussed. By the ray tracing results, it is found that the variation in the IU in the inner area of the model is less than 1% when the deflection in the outside angle of the triangle is greater than 15°, so the deflection in the outside angle of the triangle that exceeds 15° will be ignored. Then, the feature classification of the irradiance distribution is achieved by the classification principle of the inside of the triangle in the deflected 30° area and the outside of the triangle in the adjacent 15° area. Following the classification principle, the first unit has two categories, which are categorized by schematic symbols ("1-1" and "1-2") and direction arrows, as shown in the Figure 8, the second unit has four categories, and the thirst unit has two categories. The schematic symbols for the classification of the second and third units are 2-1, 2-2, 2-3, 2-4 and 3-1, 3-2, respectively, and their direction arrows are also shown in Figure 8. In other words, without considering the difference in mirror images, 16 (=2 × 4 × 2) irradiance distribution categories can be obtained.

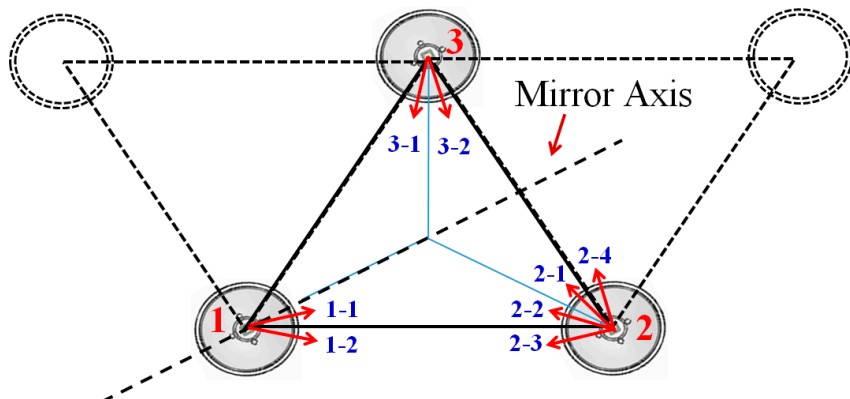

**Figure 8.** Schematic of the basic analysis model.

### 4.3. Identification of Angle Errors of the LPL Units

Next, those 16 irradiance distribution categories are regarded as classification learning targets. The task of this learning method, using the method of Haar-like feature to calculate the relative position of the maximum and minimum values for the irradiance distribution of the basic analysis model, is tried. This task is divided into the following two steps: the first step is to identify one of sixteen irradiance distribution categories and the second step is to further identify the inclination deviation value within a range.

The first step uses the so-called azimuth identification method based on Fast R-CNN learning with Haar-like features. The positions of the maximum and minimum values in the irradiance distribution images of the basic analysis model obtained by the ray tracing simulation tool are considered. By the variation in the two positions, the irradiance image can carry out the identification of 16 irradiance distribution categories. The results of identification accuracy for the first step through 10 learning epochs are listed in Table 4 and the accuracy reaches 76.03% through 10 learning epochs. The second step is the called inclination identification method. The variation in IU caused by deviation is used as the classification basis. Table 5 shows the results of identification accuracy for the deviation range of inclination angles through 16 learning epochs. One can observe that the accuracy is less stable and was 75.70% at 16 learning epochs. In other words, through the first step of classification, the deflection range of the azimuth angle for the first unit of the basic analysis model can be automatically recognized and, through the second step of classification, the deviation range of inclination angles can be automatically recognized. In addition, the second and third units of the model can also be further identified automatically. Similarly, the other units can also be identified automatically. According to this triangle model method, the angle error conditions of adjacent triangles can be calculated one by one, so as to obtain the individual angle error conditions of the twenty-three units in the simple LPL-EM.

**Table 4.** Identification accuracy results of 16 irradiance distribution categories through 10 learning epochs.

| Number of Epochs | Accuracy (%) | Number of Epochs | Accuracy (%) |
|:---:|:---:|:---:|:---:|
| 1 | 68.23 | 6 | 71.03 |
| 2 | 69.01 | 7 | 73.02 |
| 3 | 69.92 | 8 | 74.86 |
| 4 | 70.13 | 9 | 75.22 |
| 5 | 71.46 | 10 | 76.03 |

**Table 5.** Identification accuracy results for the deviation range of inclination angles through 16 learning epochs.

| Number of Epochs | Accuracy (%) | Number of Epochs | Accuracy (%) |
|---|---|---|---|
| 1 | 73.13 | 9 | 75.84 |
| 2 | 70.15 | 10 | 73.86 |
| 3 | 71.11 | 11 | 70.04 |
| 4 | 70.67 | 12 | 72.73 |
| 5 | 75.62 | 13 | 74.88 |
| 6 | 76.84 | 14 | 76.63 |
| 7 | 74.09 | 15 | 78.84 |
| 8 | 74.94 | 16 | 75.70 |

To verify the above learning method for automatic identification, the intermediate units of the simple LPL-EM are chosen. For the convenience of description, the seven LPL units have their own numbers, numbered 1 to 7, as shown in the center of Figure 4. First, the identification images of the seven LPL units with individual, specified and known angle errors are obtained by the ray tracing simulation tool. Those specified angle errors are listed in Table 6. Here, the angle errors of LPL unit No. 1 are taken as the inclination angle 1° and the azimuth angle 45° and are assumed to be known. For the triangle formed by LPL units No. 1, No. 2 and No. 3, the angle errors of LPL units No. 2 and No. 3 are obtained by the azimuth identification method and the inclination identification method, and the result is shown in Table 7. Then, for the triangle formed by LPL units No. 1, No. 3 and No. 4, the angle error of LPL unit No. 4 is obtained and shown in Table 7. Similarly, the angle errors of LPL units No. 5, No. 6 and No. 7 are obtained and shown in Table 7. The angle errors of six solved LPL units converge exactly within a particular range, that is, the automatic identification verification method is proven to be effective.

**Table 6.** Those specified angle errors of the seven LPL units for the automatic identification verification method.

| Number of LPL Unit | Inclination Angle (°) | Azimuth Angle (°) |
|---|---|---|
| 1 | 1 | 45 |
| 2 | 0.2 | 17 |
| 3 | 0.75 | 13 (Outside *) |
| 4 | 0.3 | 60 |
| 5 | 0.94 | 12 |
| 6 | 0.6 | 14 (Outside) |
| 7 | 0.9 | 56 |

* Here, "outside" refers to the angle outside the triangle.

**Table 7.** The obtained angle errors of the seven LPL units by the automatic identification verification method.

| Number of LPL Unit | Inclination Angle (°) | Azimuth Angle (°) |
|:---:|:---:|:---:|
| 2 | 0.1–0.5 | 0–30 |
| 3 | 0.6–0.8 | 0–15 (Outside *) |
| 4 | 0.1–0.5 | 30–60 |
| 5 | 0.9–1.0 | 0–30 |
| 6 | 0.6–0.8 | 0–15 (Outside) |
| 7 | 0.9–1.0 | 30–60 |

* Here, "outside" refers to the angle outside the triangle.

## 5. Conclusions

In this study, for the smart manufacturing development of PCB exposure devices, the LPL unit and LPL-EM are investigated and the angle errors of those LPL units are identified by the use of neural network learning algorithms. First, using the ray tracing simulation tool, the LPL unit with FWHM of 7.6° is designed and the LPL-EM with IU 92.9% is obtained. Then, for CNN supervised learning, the 2D irradiance distribution images, built by the ray tracing simulation tool, are used as the learning data. The variation in these learning data suggests that all units of the LPL-EM randomly have their own specific angle errors. The azimuth identification method based on Fast R-CNN learning can identify one of sixteen irradiance distribution categories and the inclination angle identification method, based on Fast R-CNN learning with Haar-like features, can obtain the inclination deviation value within a specific range. According to those results, supervised learning techniques should be able to help identify the errors of inclination and azimuth angle for the single LPL units and multi-light modules of PCB exposure devices. In other words, preliminary verification can provide a detailed understanding of the PCB exposure process of exposure devices and analyze their manufacturing problems, so as to achieve the goal of smart manufacturing.

**Author Contributions:** Conceptualization, C.-F.C. and J.-R.C.; methodology, C.-F.C.; software, J.-R.C. and T.-Y.C.; validation, C.-F.C., J.-R.C. and T.-Y.C.; writing—original draft preparation, C.-F.C.; writing—review and editing, C.-F.C. All authors have read and agreed to the published version of the manuscript.

**Funding:** This research was supported by the Ministry of Science and Technology of the Republic of China under Contract No. MOST 110-2221-E-008-064 and MOST 111-2221-E-008-037-MY3.

**Institutional Review Board Statement:** We believe this study should not require ethical approval.

**Informed Consent Statement:** We confirm this study does not involve humans.

**Data Availability Statement:** We confirm that no publicly archived datasets wave reported for this study.

**Acknowledgments:** In Acknowledgments section, you can acknowledge any support given which is not covered by the author contribution or funding sections. This may include administrative and technical support, or do-nations in kind (e.g., materials used for experiments).

**Conflicts of Interest:** The authors declare no conflict of interest.

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
