# Peer review of "Identification of the Angle Errors of the LED Parallel-Light Module in PCB Exposure Device by Using Neural Network Learning Algorithms"

_coatings, doi:10.3390/coatings12111619_

Round 1

Reviewer 1 Report

This paper presents a procedure to determine the manufacturing pointing errors of PCB exposure device made using LEDs. The procedure makes use of a convolutional neural network and the learning data has been obtained via simulation with a ray tracing simulation tool.

As a summary, the problem and prodecure are correctly formulated but some hypothesis must be reviewed (see mayor comments).

Mayor comments:

1) The procedure studies the irradiation in the center of the LPL-EM, but there are edge effects. This edge effects must be estimated.

2) Input and output variables to the CNN must be defined.

3) It seems that figure 5 corresponds to a real implementation of a LPL-EM. If so, what is the result of applying this procedure? what are the diferences between the estimated angles and the measured angles in each LPL?

4) In line 156, it is mentioned that azimuth angle is randomly selected. But what is the probability distribution of this angle deviation among LPL units?

5) Some experimental results must be included to validate the procedure.

Minor comments:

1) A drawing of a single LPL unit and the error angle range must be justified.

2) English language shall be reviewed. as an example, it must be "are ignored" in line 113 and "variation" in line 117.

3) Use "irradiated pixels" instead of "irradiance pixels" anywhere in the text.

4) Definition of IU shall be in line 84.

5) Please, define quadrant in table 1

6) Please, define Outside in table 6 and 7

Author Response

Dear reviewer,

Thank you for your time and consideration of our work. And thank the referees for a thorough and helpful analysis of our manuscript. We have individually responded to all suggestions and comments from the referees and revised the manuscript using the “Track Changes” function accordingly. In addition, we have checked all references to confirm manuscripts relevant to the content of the article, and removed four references that were less relevant. In the revised manuscript, we use a combination of red words and yellow fluorescent areas to remind these revisions. The response letter is attached.

Reviewer 2 Report

Dear Authors, please find my comments regarding your paper: Identification for the Angle Errors of the LED parallel-light module in PCB Exposure Device by Using Neural Network Learning Algorithms, in Coatings.

C0: First of all, please define "PCB exposure" in the abstract.

C1: "Obviously, the supervised learning technology...
Please avoid such comments, and take care, that this is found in the abstract. If something is obvious, then what is a motivation behind the proper research?

C2: Please use sans-serif fonts in the paper's figures to improve clarity. Please use vector figures, e.g. Fig 7 seems to be distorted and overcompressed.

C3: Please give proper units everywhere, where it is necessary- (Irradiance, for example -> W⋅m−2 ?)

C4: Please define resolution according to Figure 5 -> the grid seems to be 25x25, the actual range is 150-150 mm, what was the limit/consideration behind the current setup? Please give also consideration behind 23 LPL units to clarify the resolution limitations.

C5: Is accuracy around 70-80% enough? Please elaborate.

CX: There are minor spelling errors, only a spellcheck is required on the paper - otherwise English and text flow is good. Text is dense at given chapters, i'd recommend to use subchapters to improve the readability a bit.

Author Response

(The authors gave the same response as above.)

Round 2

Reviewer 1 Report

The authors have appropriately updated the first draft and it is now valid for publication.

Reviewer 2 Report

accept